# High output mode-locked laser empowered by defect regulation in 2D Bi$_2$O$_2$Se saturable absorber

Junting Liu [1,3], Fang Yang[2,3], Junpeng Lu [2✉], Shuai Ye[1], Haowen Guo[1], Hongkun Nie[1], Jialin Zhang[2], Jingliang He[1], Baitao Zhang [1✉] & Zhenhua Ni [2✉]

Atomically thin Bi$_2$O$_2$Se has emerged as a novel two-dimensional (2D) material with an ultrabroadband nonlinear optical response, high carrier mobility and excellent air stability, showing great potential for the realization of optical modulators. Here, we demonstrate a femtosecond solid-state laser at 1.0 μm with Bi$_2$O$_2$Se nanoplates as a saturable absorber (SA). Upon further defect regulation in 2D Bi$_2$O$_2$Se, the average power of the mode-locked laser is improved from 421 mW to 665 mW, while the pulse width is decreased from 587 fs to 266 fs. Moderate Ar$^+$ plasma treatments are employed to precisely regulate the O and Se defect states in Bi$_2$O$_2$Se nanoplates. Nondegenerate pump-probe measurements show that defect engineering effectively accelerates the trapping rate and defect-assisted Auger recombination rate of photocarriers. The saturation intensity is improved from 3.6 ± 0.2 to 12.8 ± 0.6 MW cm$^{-2}$ after the optimized defect regulation. The enhanced saturable absorption and ultrafast carrier lifetime endow the high-performance mode-locked laser with both large output power and short pulse duration.

[1] State Key Laboratory of Crystal Materials, Institute of Novel Semiconductors, Shandong University, 250100 Jinan, Shandong, China. [2] School of Physics and Key Laboratory of MEMS of the Ministry of Education, Southeast University, Nanjing 211189, China. [3]These authors contributed equally: Junting Liu, Fang Yang. ✉email: phyljp@seu.edu.cn; btzhang@sdu.edu.cn; zhni@seu.edu.cn

Ultrashort pulsed lasers with high output power have been proven indispensable in many applications, such as biomedicine[1], spectroscopy[2] and precision micro/nanoprocessing[3]. Saturable absorber (SA) is the crucial optical element to trigger pulsed operations. It can switch between different absorption states on an ultrafast timescale and generate ultrafast lasers[4]. Currently, the semiconductor saturable absorber mirror (SESAM) is the most commonly used SA[5]. However, SESAM requires sophisticated fabrication and exhibits narrow nonlinear optical bandwidths, which results in a narrow wavelength tuning range[6]. 2D materials, such as graphene[7,8], transition metal dichalcogenides (TMDs; e.g., $MoS_2$ and $WS_2$)[9–12], MXene[13] and topological insulators (TIs; e.g., $Bi_2Se_3$ and $Sb_2Te_3$)[14], have been demonstrated to present broadband response wavelengths, while flexibility and atomic smooth surfaces make them easy to integrate in device fabrication. This renders 2D materials promising candidates for SA in the development of pulsed lasers[15,16]. However, the weak absorptance (single layer: ~2.3%) and limited density of states in graphene are obstacles to achieving a high modulation depth. The extremely high saturation intensity of TMDs on the order of tens of GW cm$^{-2}$ makes continuous-wave mode-locked (CWML) operation difficult in all-solid-state lasers[17,18]. In addition, the complex fabrication process and poor air stability limit the application of MXene and TIs materials in pulsed lasers[19,20].

Recently, 2D bismuth oxyselenide ($Bi_2O_2Se$) has been demonstrated to have ultrabroadband nonlinear modulation (from 1.55 μm to 5.0 μm) and a larger nonlinear absorption coefficient β ($-2.91 \times 10^{-6}$ cm W$^{-1}$ @800 nm)[21]. In addition, the excellent air stability and plain fabrication process of $Bi_2O_2Se$ are conducive to improving the stability of pulsed laser devices[22]. However, ultrafast lasers based on $Bi_2O_2Se$ SA have not been devised. This is probably due to the low saturation intensity of $Bi_2O_2Se$ (~ MW cm$^{-2}$)[23], which easily bleaches in fiber mode-locked lasers. Compared with fiber lasers, SAs with low saturation intensity are more suitable for all-solid-state lasers with low intracavity energy[4]. Therefore, it is urgent to explore the capacity of $Bi_2O_2Se$ SA to generate ultrafast pulses in all-solid-state lasers. More importantly, the longer carrier lifetime (~ 200 ps)[24] will significantly hinder the realization of mode-locked lasers with high output and ultrashort pulses based on 2D $Bi_2O_2Se$ SA. Therefore, a straightforward and effective approach to directly modulate the carrier dynamics and nonlinear absorption properties in 2D $Bi_2O_2Se$ SA is desired.

In this work, we demonstrate a femtosecond solid-state laser (λ ~ 1.0 μm) with $Bi_2O_2Se$ nanoplates as SA. A high output power of ~665 mW and an ultrashort pulse width of ~266 fs are achieved upon defect engineering in the 2D $Bi_2O_2Se$ SA. Moderate Ar$^+$ plasma generated from a uniplanar plasma generator is employed to precisely regulate the O and Se defect states in $Bi_2O_2Se$ nanoplates. The nondegenerate pump-probe measurements show that defect regulation effectively accelerates the trapping rate and defect-assisted Auger recombination rate of photocarriers. The maximum trapping/recombination rates reach ~0.1 cm$^2$ s$^{-1}$ and $\sim 4.82 \times 10^{-17}$ cm$^4$ s$^{-1}$, respectively. Defect regulation also triggers strong saturable absorption and self-defocusing properties in $Bi_2O_2Se$ in the near-infrared region. The saturation intensity is greatly improved from ~3.6 ± 0.2 to ~ 12.8 ± 0.6 MW cm$^{-2}$ after the optimized defect regulation. The improved saturation intensity and ultrafast carrier lifetime endow the high-performance mode-locked laser with both high output and short pulse duration.

## Results

### Pulse laser and photocarrier lifetime modulation. The $Bi_2O_2Se$ nanoplates are synthesized via a CVD approach. The detailed

characterizations of the products are shown in Supplementary Fig. 1. The as-grown $Bi_2O_2Se$ nanoplates are transferred onto a mirror with a high-reflection (HR) coating at 1020–1100 nm as a saturable absorption mirror (SAM). With the $Bi_2O_2Se$ SAM in the resonator (Supplementary Fig. 2a), stable continuous-wave mode-locked (CWML) operation is achieved when the absorbed pump power exceeds 5.80 W. The maximum average output power of ~421 mW is obtained when the absorbed pump power is increased to 7.82 W (Fig. 1a). The pulse duration is extracted to be ~587 fs from Sech$^2$ pulse shape fitting (Fig. 1b). The spectrum of a CWML laser is centered at 1047.0 nm with a full width at half-maximum (FWHM) of 2.1 nm (Supplementary Fig. 2b). The CWML pulse trains at the maximum average output power show good amplitude stability, as recorded with time spans of 500 ns (Supplementary Fig. 2c) and 1 μs (Supplementary Fig. 2d). The radio frequency (RF) spectrum in a wider span from 0 to 1 GHz without spurious frequency components or modulations shows a resolution bandwidth (RBW) of 7 kHz (insert of Fig. 1c). The signal-to-noise ratio for the fundamental peak located at 42.35 MHz is measured to be ~55 dB (Fig. 1c). The absence of any spurious modulations and high stability implies the clean CWML operation of the $Bi_2O_2Se$ SA.

The implementation of mode locking and optimization of the output power and pulse width are determined by the saturation intensity and recovery time of SA. The saturation intensity can typically be controlled by adjusting the recovery time. A lower saturation intensity generates Q-switched mode locking and limited output power, while an exorbitant saturation intensity leads to an excessively high threshold power for mode locking[5]. Moreover, it should be noted that longer lifetime decay channels usually have a greater leading role than fast decay channels, especially in the initial pulse formation phase, which will directly determine the pulse width of the mode-locked laser[4]. In this case, various strategies have been proposed to tune the slow component of the recovery time of SESAMs[25,26]. Therefore, to optimize the performance of lasers with higher output power and shorter pulse widths, both the saturation intensity and carrier lifetime of SA should be precisely controllable and adjustable. Considering the regulable density of states of 2D materials, defect engineering via plasma irradiation is employed to optimize the saturation intensity and carrier lifetime of $Bi_2O_2Se$ SA.

The modulated photocarrier dynamics of 2D $Bi_2O_2Se$ are investigated by nondegenerate pump-probe spectroscopy. A schematic illustration of the experimental setup is shown in Supplementary Fig. 3a. Pulses with a photon energy of 3.1 eV are used to excite photocarriers, while the differential transmission (ΔT/T) of the time-delayed 1.83 eV pulses is used to probe the dynamics (Supplementary Fig. 3b). Photoinduced bleaching (positive ΔT/T under different pump fluences (F) (from 17.5 μJ cm$^{-2}$ to 71.9 μJ cm$^{-2}$) is observed in pristine $Bi_2O_2Se$ (Supplementary Fig. 4a). Bleaching arises from the filling of empty states and Pauli blocking by photocarriers[27]. The transient bleaching processes are described well by a biexponential model[28]. The best-fitting parameters are listed in Supplementary Table 1. Evidently, both $\tau_1$ (~1 ps) and $\tau_2$ (~100 ps) are independent of pump fluences, while the proportion of fast processes ($A_1$) decreases with increasing pump fluences. In addition, the decay processes also show temperature (T) independence in the range of 80–300 K (Supplementary Fig. 4b). Therefore, the carrier cooling process that usually occurs on an ~0.5-2 ps time scale via carrier-phonon scattering is excluded because it is temperature- and pump fluence-dependent[29]. The fast decay is attributed to the defect trapping of carriers[30,31]. The occupation of defect states is saturated at high carrier density, causing a decrease in the proportion of the fast decay process.

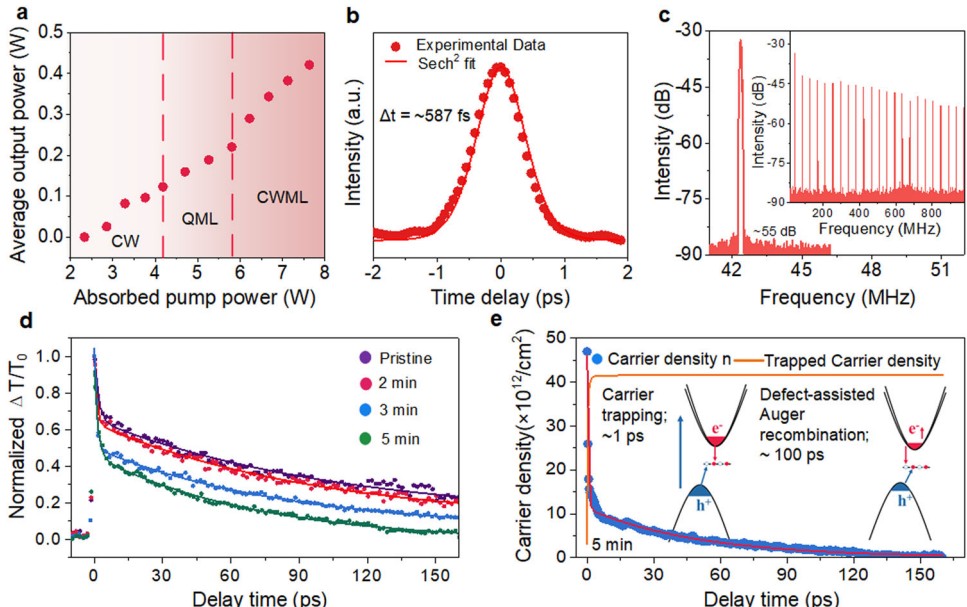

**Fig. 1 Mode-locked laser based on pristine Bi$_2$O$_2$Se nanoplates. a** Average output power of the CWML versus absorbed pump power. As the pump power increases, the mode-locked operating state transitions from continuous wave (CW) to Q-switched mode-locking (QML), and finally switches to CWML state. **b** Autocorrelation trace for ~587 fs duration. **c** Recorded frequency spectrum with narrow and wide (inset) spans. **d** Normalized differential transmission of Bi$_2$O$_2$Se nanoplates with different plasma treatment times at T = 300 K and F = 71.9 µJ/cm$^2$. **e** Blue dots are the time evolution of the carrier density. The red (orange) line indicates the simulated temporal evolution of the free (trapped) carrier density using Eq. (1)(2). Inset: Schematic illustration of carrier trapping by defects and the subsequent defect-assisted Auger recombination.

**Table 1 The fitting parameters extracted from Eqs. (1) and (2) to model pump-probe traces of Bi$_2$O$_2$Se nanoplates under different plasma irradiation times.**

| Irradiation time (min) | D1 | D2 | $\tau_1$ (ps) | $\tau_2$ (ps) | C (×10$^{-17}$ cm$^4$ s$^{-1}$) | $n_d$ (×10$^{13}$ cm$^{-2}$) | $\gamma$ (×10$^{-2}$ cm$^2$ s$^{-1}$) |
|---|---|---|---|---|---|---|---|
| 0 | 0.33 | 0.67 | 0.56 ± 0.06 | 106.2 ± 12.2 | 1.58 | 2.95 | 3.91 |
| 2 | 0.42 | 0.58 | 0.91 ± 0.06 | 98.4 ± 1.7 | 1.85 | 3.14 | 4.77 |
| 3 | 0.45 | 0.55 | 1.12 ± 0.05 | 89.1 ± 3.6 | 2.57 | 3.41 | 5.68 |
| 5 | 0.52 | 0.48 | 1.07 ± 0.04 | 51.2 ± 0.8 | 4.82 | 4.17 | 10.41 |

The regulation of the carrier dynamics is obtained by controlling the irradiation time (0 min, 2 min, 3 min, and 5 min) of Ar$^+$ plasma (Fig. 1d). The best-fitting parameters of the decay curves are listed in Table 1. The lifetime ($\tau_2$) and proportion ($A_2$) of the slow process decrease with increasing irradiation time, while the fast process becomes more significant. The increase in $A_1$ is probably due to the increase in defect density with increasing irradiation time. The slow decay arises from defect-assisted Auger recombination after the carrier is captured at the defect sites (inset of Fig. 1e)[31,32]. In general, the decay rate via Auger recombination is temperature-independent, and it becomes faster with increasing defect density[33]. The decay rate and coefficient of different processes can be extracted from the coupling rate equations[30],

$$\frac{dn}{dt} = -\gamma(n_d - n_{tr})n - Cn_{tr}n^2 \qquad (1)$$

$$\frac{dn_{tr}}{dt} = \gamma(n_d - n_{tr})n - Cn_{tr}n^2 \qquad (2)$$

where $n$ is the carrier density, $n_d$ is the defect density, $\gamma$ is the carrier trapping coefficient, $n_{tr}$ is the density of trapped carriers, and $C$ is the Auger coefficient. The dynamic evaluation of $n$ and $n_{tr}$ of 2D Bi$_2$O$_2$Se irradiated for different durations is depicted in Fig. 1e and Supplementary Fig. 4d–f. $n_d$, $C$ and $\gamma$ are extracted and shown in Table 1. Evidently, the carrier capture rate ($\gamma$),

Auger recombination rate ($C$), and defect density ($n_d$) significantly increase with increasing irradiation time. With 5-min plasma treatment, $\gamma$ and $n_d$ reach ~0.1 cm$^2$ s$^{-1}$ and ~4.17 × 10$^{13}$ cm$^{-2}$, respectively. The resulting defect-assisted Auger recombination rate is ~4.82 × 10$^{-17}$ cm$^4$ s$^{-1}$. This indicates the success of the regulation of carrier dynamics in 2D Bi$_2$O$_2$Se via plasma treatment, and the regulation is probably caused by the modulation of defect states.

**TEM analysis of Bi$_2$O$_2$Se nanoplates**. To identify the modulation of defects in 2D Bi$_2$O$_2$Se, high-resolution TEM (HRTEM) is employed to investigate the effects of plasma treatments at the atomic scale. Figure 2a–c shows the typical large-area HRTEM image, selected area electron diffraction (SAED) pattern and close-up TEM image of pristine Bi$_2$O$_2$Se nanoplates, respectively. High quality of the sample is indicated, and the d-spacings of 0.27 nm are assigned to the {110} planes. The abovementioned characterization is also performed on the Bi$_2$O$_2$Se nanoplates after short-term (2 min, Fig. 2d–f) and long-term (5 min, Fig. 2g–i) argon plasma treatment. Evidently, new diffraction spots, as highlighted by the yellow circles with weak intensity, emerge after short-term plasma treatment (Fig. 2e). These spots could be assigned to the newly formed polycrystalline {020} planes upon plasma treatment, which is also confirmed by the d-spacing of 0.20 nm in Fig. 2f. After long-term plasma treatment, the intensity of the diffraction spots from polycrystalline

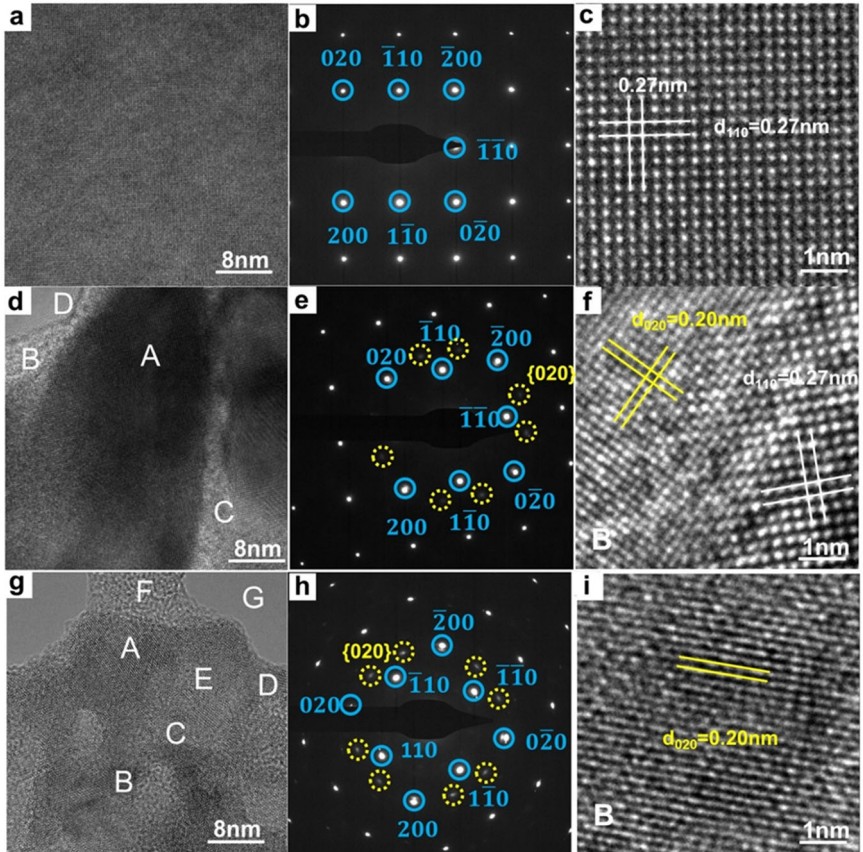

**Fig. 2 TEM analysis of Bi₂O₂Se nanoplates. a** Large-scale TEM image, **b** corresponding SAED pattern marked by blue circles and **c** close-up TEM image of pristine Bi₂O₂Se. The d-spacing of 0.27 nm (marked by white lines) can be assigned to the {110} planes. **d** Large-scale TEM image, **e** corresponding SAED pattern (the newly formed polycrystalline {020} planes are marked by yellow circles), and **f** close-up STM images of region B highlighted in panel **d** of the Bi₂O₂Se sample after short-term plasma treatment. **g** Large-scale TEM image, **h** corresponding SAED pattern and **i** close-up STM images of region B highlighted in panel **g** of the Bi₂O₂Se sample after long-term plasma treatment. The d-spacings of 0.27 nm and 0.20 nm (marked by yellow lines) can be assigned to the {110} planes and {020} planes, respectively.

{020} planes increased (Fig. 2h). More regions of the surface transform to structures with d-spacings of 0.20 nm, and domains with different orientations are observed. Some areas even turn into disordered structures. More HRTEM images of the Bi₂O₂Se sample subjected to different plasma treatment durations are shown in Supplementary Fig. 5a–i. These images reveal d-spacings of 0.20 and 0.27 nm, which correspond to the crystal planes of {020} and {110} of tetragonal Bi₂O₂Se, respectively, according to the powder diffraction file PDF#70-1549. From this, it is conjectured that argon plasma treatment is an effective method to trigger defect engineering in Bi₂O₂Se nanoplates[34,35]. The elemental distribution of Bi₂O₂Se nanoplates was analyzed by X-ray spectroscopy (EDS), as shown in Supplementary Fig. 6d–m. Supplementary Fig. 1g shows the EDS spectra obtained from the pristine, short- and long-term argon plasma-treated Bi₂O₂Se nanoplates, corresponding to the high-angle annular dark-field (HAADF) image in Supplementary Fig. 6a–c. The intensities of the O peaks in the pristine Bi₂O₂Se nanoplates at 524 eV are lower than those of the others. With longer argon plasma treatment times, the Se and Bi atom contents decrease from 38.6% to 21.6% and 23.2% to 14.6%, respectively, while the O atom content increases from 38.2% to 63.7% (Supplementary Fig. 1h).

**Defect-mediated nonlinear absorption**. To investigate the effect of defect engineering on the NLO responses of Bi₂O₂Se nanoplates, the open-aperture (OA), closed-aperture (CA) Z-scan and I-scan methods are performed with homemade picosecond fiber

lasers operating at 1064 nm (details in the Methods). Figure 3a presents the OA results of Bi₂O₂Se nanoplates under different plasma irradiation times with an input pulse energy of 0.498 μJ. The symmetrical peak at the focus (z=0) indicates that the light transmittance increases with increasing laser beam intensity (the sample moves to the focus), arising from the saturable absorption effect of the Bi₂O₂Se nanoplates. To quantitatively evaluate the nonlinear absorption coefficients of Bi₂O₂Se nanoplates, the nonlinear absorption coefficients ($\beta_{eff}$) and the imaginary part of the third-order nonlinear susceptibility $Im\chi^{(3)}$ can be fitted by Supplementary Eqs. (1, 2). All of the fitted $\beta_{eff}$ and calculated $Im\chi^{(3)}$ of Bi₂O₂Se nanoplates are summarized in Supplementary Table 2. Note that the nonlinear absorption coefficient $\beta_{eff}$ is negative and the absolute value decreases as the plasma irradiation time increases, which is mainly due to the increase in defect density resulting in large scattering loss[36]. Similarly, the calculated $Im\chi^{(3)}$ also experiences a similar trend as $\beta_{eff}$. The maximum values of $\beta_{eff}$ and $Im\chi^{(3)}$ are $(-437 \pm 5)$ cm MW$^{-1}$ and $(-3.36 \pm 0.04) \times 10^{-6}$ esu, respectively, which are significantly larger than those of other NLO materials (as shown in Table 2). In addition, the curve of the nonlinear transmission with respect to the incident laser intensity can be obtained by the I-scan method (as shown in Fig. 3b). The curves can be well fitted by an SA model of one-photon absorption[37].

$$T(Z) = 1 - \frac{\triangle R}{1 + I/I_s} - A_{ns} \qquad (3)$$

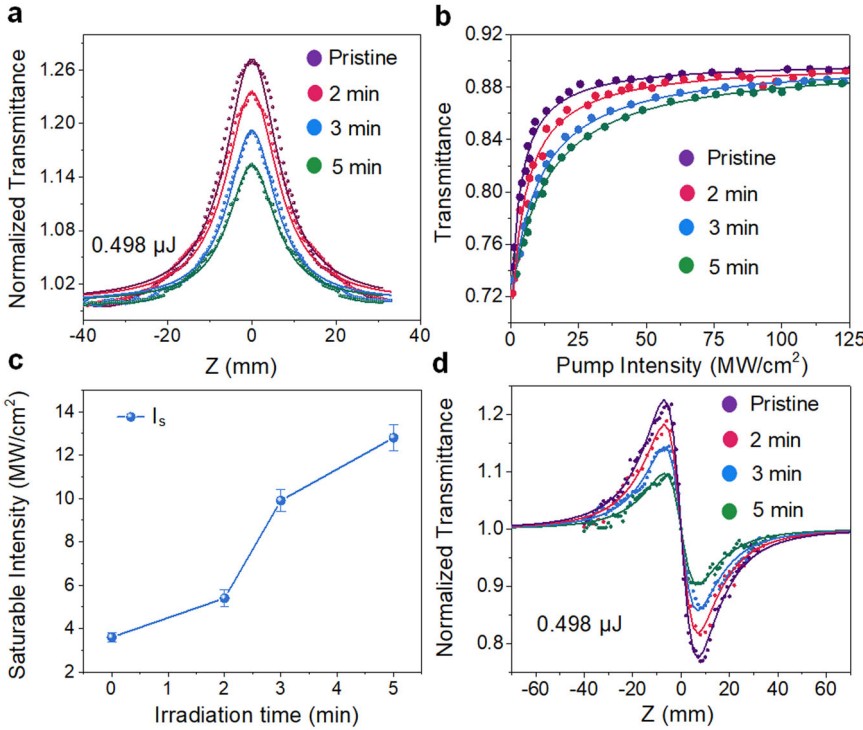

**Fig. 3 Defect-mediated nonlinear absorption properties. a** Open-aperture Z-scan results of the $Bi_2O_2Se$ nanoplates under different plasma irradiation times with an input pulse energy of 0.498 μJ. It can be well fitted by Supplementary Equation (1) (solid lines). **b** Nonlinear transmittance of $Bi_2O_2Se$ nanoplates under different plasma irradiation times. The solid lines show that the fitting results by Eq. (3) are well-matched with the experimental results. **c** Trend of the saturation intensity of $Bi_2O_2Se$ nanoplates under different plasma irradiation times. The error bars are the fitting error, which are obtained from the fitting results by Eq. (3). **d** Closed-aperture Z-scan results of the $Bi_2O_2Se$ nanoplates under different plasma irradiation times with an input pulse energy of 0.498 μJ. It can be well fitted by Supplementary Equation (4) (solid lines).

**Table 2 Comparison of $\beta_{eff}$ and $n_2$ values between $Bi_2O_2Se$ nanoplates and other nanomaterials.**

| Materials | Laser parameters | $\beta_{eff}$ (cm $GW^{-1}$) | $n_2$ ($m^2$ $W^{-1}$) | Reference |
|---|---|---|---|---|
| Graphene | 1030 nm, 1 kHz, 340 fs | $-(19.27 \pm 0.89) \times 10^{-2}$ | $-13.7 \times 10^{-16}$ | 41 |
| $MoS_2$ | 800 nm, 1 kHz, 130 fs | $-(8.05 \pm 0.37) \times 10^{-3}$ | $-(0.907 \pm 0.001) \times 10^{-19}$ | 44 |
| $WS_2$ | 1064 nm, 20 Hz, 25 ps | $-5.1 \pm 0.26$ | $(5.83 \pm 0.18) \times 10^{-15}$ | 42 |
| BP | 800 nm, 1 kHz, 100 fs | $-(6.82 \pm 0.12) \times 10^{-2}$ | $-(3.61 \pm 0.19) \times 10^{-16}$ | 43 |
| MXene | 1064 nm, 1 kHz, 100 fs | $-0.206$ | $-3.4\ 7 \times 10^{-20}$ | 45 |
| GeP | 800 nm, 1 kHz, 100 fs | $-0.474$ | $2.09 \times 10^{-18}$ | 39 |
| $CH_3NH_3PbI_3$ film | 1064 nm, 40 Hz, 40 ps | $-2.3 \times 10^3$ | $3.7 \times 10^{-15}$ | 46 |
| $Bi_2O_2Se$ | 1064 nm, 300 kHz, 10 ps | $-(4.37 \pm 0.05) \times 10^5$ | $-(5.29 \pm 0.03) \times 10^{-13}$ | This work |

where $I_s$ is the saturation intensity, $\Delta R$ is the modulation depth, and $A_{ns}$ is the nonsaturable loss. As shown in Supplementary Table 2, the values of $I_s$, $\Delta R$, and $A_{ns}$ for $Bi_2O_2Se$ nanoplates under different irradiation times are obtained by fitting the experimental data. Obviously, as shown in Fig. 3c, the value of the saturation intensity ($I_s$) shows an increasing trend as the plasma irradiation time increases (from $3.6 \pm 0.2$ MW $cm^{-2}$ to $12.8 \pm 0.6$ MW $cm^{-2}$), while the value of the modulation depth ($\Delta R$:~19%) and nonsaturable loss ($A_{ns}$: ~10%) basically remain unchanged. This may be because of defect engineering in $Bi_2O_2Se$ nanoplates, which accelerates the process of photocarrier dynamics, leading to an increasing tendency of saturation intensity.

The CA Z-scan technique is used to characterize the nonlinear refractive index, which is an important parameter for quantifying the Kerr nonlinearity of an NLO material[38]. The normalized transmittance of CA Z-scan results under different plasma irradiation times at a wavelength of 1064 nm of $Bi_2O_2Se$ nanoplates is shown in Fig. 3d, which can be fitted by the following formula[39,40]:

$$T = 1 + \frac{4kL_{eff}n_2I_0z}{z_0(z^2/z_0^2 + 9)(z^2/z_0^2 + 1)} \quad (4)$$

where $k = 2\pi/\lambda$ and $n_2$ is the nonlinear refractive index. The peak-valley shapes of the normalized transmittance CA/OA Z-scan curves suggest negative refractive indices, which result from the Kerr effect-induced self-defocusing effect in $Bi_2O_2Se$ nanoplates. The largest values of $n_2$ of pristine $Bi_2O_2Se$ nanoplates are determined to be $-(5.29 \pm 0.03) \times 10^{-13}$ $m^2$ $W^{-1}$, and the resulting nonlinear refractive indices show a similar trend as that of $\beta_{eff}$ (Supplementary Fig. 7). As shown in Table 2, the value of $n_2$ is much higher than that of most 2D materials[41–46]. In general, the large values of $\beta_{eff}$ and $n_2$ illustrate the strong light interaction, indicating the enormous potential applications of $Bi_2O_2Se$ nanoplates for optical modulators. Moreover, defect engineering in $Bi_2O_2Se$ nanoplates flexibly adjusts its nonlinear absorption parameter, including the saturation influence and saturation recovery time, which are the key parameters for laser pulse

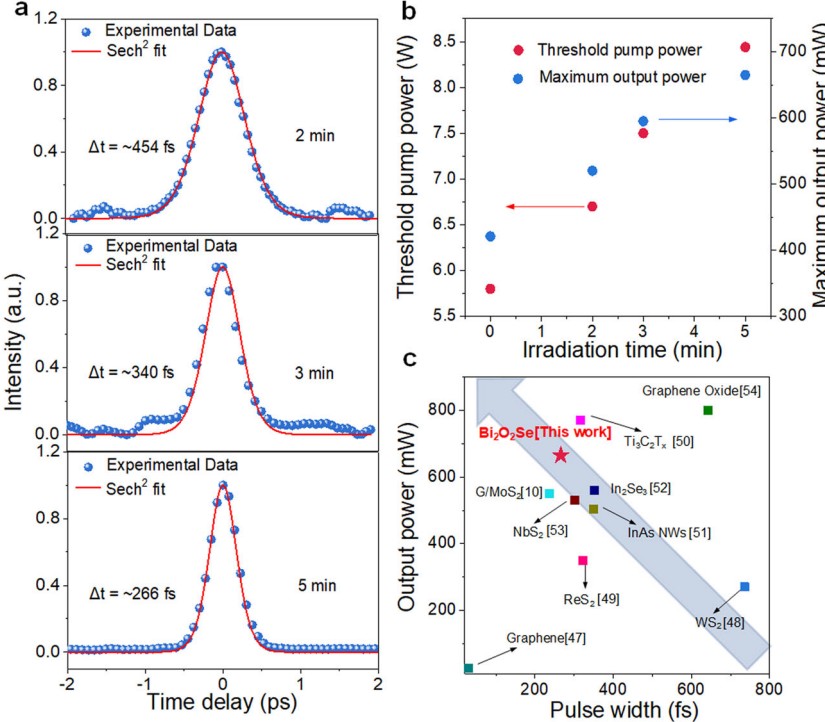

**Fig. 4 Mode-locked laser results based on $Bi_2O_2Se$ nanoplates under different plasma irradiation times. a** Autocorrelation trace under different plasma irradiation times. **b** Trend of the maximum average output power and threshold pump power of the CWML based on $Bi_2O_2Se$ SA treated with different argon plasma irradiation times. **c** Results of mode-locked solid-state bulk lasers operating at 1.0 μm based on various nanomaterials.

generation. Therefore, the demonstrated good NLO response promises optoelectronic applications of $Bi_2O_2Se$ nanoplates.

**The Improvement of Pulsed Laser Performance.** The photocarrier relaxation process and nonlinear absorption properties of $Bi_2O_2Se$ nanoplates can be flexibly controlled by defect engineering, which prompts us to further explore their capability for ultrashort pulse generation. As shown in Fig. 4a, the narrowest pulse duration ($\Delta t$) is measured to be 266 fs by $Sech^2$ pulse shape fitting, and the corresponding FWHM of the spectrum is 4.8 nm (shown in Supplementary Fig. 8c) based on the $Bi_2O_2Se$ nanoplate SAM, which is treated with argon plasma for 5 mins. The time bandwidth of 0.349 indicates that the output pulse is a little chirped. The spectrum of the mode-locked operation based on $Bi_2O_2Se$ nanoplates under 2 min and 3 min plasma irradiation times is shown in Supplementary Fig. 8a, b. Moreover, the generation threshold and maximum average output power of CWML operation increase with increasing plasma irradiation time on the $Bi_2O_2Se$ nanoplate SAM, and the maximum reaches 8.44 W and 665 mW, respectively (Fig. 4b). This is mainly because the increase in oxygen vacancy defects increases the saturation influence of the $Bi_2O_2Se$ nanoplates, which is more difficult to saturate and more suitable for working in high-power mode-locked oscillators[5]. While further increasing the pump power, the $Bi_2O_2Se$ SA will be completely bleached, and there will be insufficient further modulation to drive the pulse forming process. Eventually, the CWML operation became unstable and disappeared. Notably, no optical damage is observed in the $Bi_2O_2Se$ SAs during the mode-locking operation. The instabilities (average output power, rms) of CWML operation are measured to be 1.69% at 12 h (Supplementary Fig. 9a), and the characterization of CWML operation (Supplementary Fig. 9b, c) for the same $Bi_2O_2Se$ SA treated with argon plasma for 5 mins before and after exposure to air for three months demonstrates that the device has excellent long-term stability. From the summarization

of the femtosecond mode-locked solid-state bulk laser based on different nanomaterials (Fig. 4c)[10,47–54], it is obvious that the pulsed laser with $Bi_2O_2Se$ nanoplates has a relatively shorter pulse width and higher average output power. In addition, $Bi_2O_2Se$ SA treated with argon plasma for 5 mins was employed to generate mode-locked pulses at 2.0 μm. As shown in Supplementary Fig. 10, the good performance of the mode-locked laser at 2.0 μm reveals the excellent broadband modulation properties of $Bi_2O_2Se$ SA, which has the potential to be a mid-infrared star material (the details are shown in Supplementary VII). In general, the precise control of the photocarrier dynamics and nonlinear absorption properties of $Bi_2O_2Se$ nanoplates via defect engineering improves the average power of its corresponding mode-locked laser, which is of great significance to the realization of high-power femtosecond pulse output.

## Discussion

In summary, we have demonstrated a near-infrared (NIR, λ ~1.0 μm) femtosecond solid-state laser with $Bi_2O_2Se$ nanoplates as the SA. Upon further defect engineering in the 2D $Bi_2O_2Se$ SA, a high output power (~665 mW) and ultrashort pulse width (~266 fs) are achieved. Defect engineering is triggered by the regulation of the O and Se defect states in $Bi_2O_2Se$ nanoplates via moderate $Ar^+$ plasma treatments. Defect regulation effectively accelerates the trapping rate and defect-assisted Auger recombination rate of photocarriers. Defect regulation also empowers strong saturable absorption and self-defocusing properties in $Bi_2O_2Se$ in the NIR region. The improved saturation intensity and ultrafast carrier lifetime make synergetic contributions to the high-performance mode-locked laser with both high output and ultrashort pulse width.

## Methods

**Sample growth.** Few-layer $Bi_2O_2Se$ was synthesized by the chemical vapor deposition (CVD) method. The $Bi_2Se_3$ and $Bi_2O_3$ powders were placed in the high-

temperature zone for the generation of vapor sources by sublimation, while the micrasubstrate was placed ~15 cm downstream of the center. Under a pressure of 100 Torr, the growth temperature was elevated to 680 °C in 25 min and held for 30 min, and the flow rate of the carrier gas argon was 150 sccm.

**Nondegenerate micro pump-probe measurements**. We established a nondegenerate micropump-probe setup by using Ti:sapphire oscillators (800 nm, 80 MHz, 150 fs), which were separated into two components. One beam was fixed at 400 nm and was used as the pump light. The other beam was used to drive the optical parametric oscillator to generate pulses from 1000 to 1500 nm. After frequency doubling through the BBO crystal, the wavelength of the beam was changed to 500–750 nm. The wavelength of 650 nm was used as the probe light. Both the pump and probe light were focused onto the sample by a 20× objective lens, and the pump fluence was 20 times higher than the probe fluence. A CCD camera (Thorlabs, DCC1545 M) was used to image the sample and to check that the center of the pump and probe light coincided. The chopper frequency was fixed to 1 kHz as the reference frequency of the lock-in amplifier. The pump-induced change in the probe was detected by an adjustable gain balanced photoreceiver (Newport, 2317NF).

**Z-scan and I-scan measurements**. A homemade mode-locked Yb fiber laser (center wavelength 1064 nm, repetition rate 100 kHz-1 MHz, pulse duration 10 ps) was used to perform the Z-scan and I-scan measurements. In the Z-scan setup, the pulses were divided into two parts: one part was set as a reference light, collected by a power meter (Thorlabs S470C); the other part was focused by a lens ($f = 100$ mm) into the samples, and the focused beam waist was estimated to be 52 μm. The sample was fixed to a stepper motor, which was controlled by a computer program. In the Z-scan setup, the pulses were also divided into two parts: one part was set as a reference light, the other part was focused by a lens ($f = 125$ mm) into the samples, and the focused beam waist was estimated to be 65 μm. The optical power density on the sample is changed by the attenuator.

**CW mode-locking operation at 1.0 μm**. $Yb^{3+}$-ion-doped (10 atom %) Yb:KYW crystal ($3 \times 3 \times 4$ mm) was used as the gain medium and was wrapped in indium foil and cooled by running water at a temperature of 18 °C. The pump source was a 27 W fiber-coupled laser diode (numerical aperture of 0.22, core diameter of 105 μm) emitting at 976 nm, which was focused into the laser gain medium via a 1:1 coupling optics system well matched with the radius of the $TEM_{00}$ cavity mode (57 μm) calculated by ABCD propagation matrix theory, and the oscillation laser mode radius on $Bi_2O_2Se$ SA was 35 μm. A plane OC with a transmittance of 1% for the spectral range of 1000–1100 nm was used. To compensate for the intracavity group delay dispersion (GDD), we use a two-plane Gires-Tournois interferometer (GTI) mirror to provide a total GDD of $\approx -750$ fs$^2$ per round.

## Data availability

Relevant data supporting the key findings of this study are available within the article and the Supplementary Information file. All raw data generated during the current study are available from the corresponding authors upon request. Source data are provided with this paper.

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

## Acknowledgements

We sincerely appreciate Prof. Chaokuei Lee from National Sun Yat-Sen University, Prof. Jie Ma from Jiangsu Normal University and Prof. Weidong Chen from Fujian Institute of Research on the Structure of Matter for discussing the obtained mode-locking pulse shapes. This work was supported by the National Research Foundation of China (Grant No. 61975095, 61927808, 62174026, 61975097, 91963130), the National KRDPC (2019YFA0308000, 2017YFA0205700), the Youth Cross Innovation Group of Shandong University (Grant No. 2020QNQT), and the Financial Support from Qilu Young Scholar of Shandong University.

## Author contributions

J.L., B.Z. and Z.N. conceived the project. J.L. carried out the experiments. F.Y. prepared the materials. S.Y., H.N. and H.G. helped with the laser experiments. J.Z. performed the TEM characterization. J.H. leads the project.

## Competing interests

The authors declare no competing interests.
