## [Peer Review File · Nature Communications]

High output mode-locked laser empowered by defect regulation in 2D Bi₂O₂Se saturable absorberREVIEWER COMMENTS

Reviewer #1 (Remarks to the Author):

This work demonstrates the availability of mode-locking performance controlling using Bi₂O₂Se nanoplates as a saturable absorber (SA) via defect engineering (i.e. plasma irradiation), which enables tailoring of the saturation intensity and carrier lifetime. An improved output power and narrowed pulse width of mode-locking have been achieved based on the optimized SA. The detailed characterizations of both the material and laser have been also performed for intuitive comparisons. This approach proposed in the paper is new and should be quite valuable for the development of ultrafast laser field. In my opinion, this work can be considered to be published, after addressing the following issues:

- (1) The achieved mode-locked pulses are not operated at soliton state, why sech² instead of Gauss profile was selected to fit the autocorrelation trace?
- (2) How about the long-term stability of the mode-locked laser?
- (3) What limit further power scaling of the mode-locked pulses? Damage of the SA, disappearance of mode-locking, or others? The relative comments should be provided.
- (4) Following the above question, whether the mode-locking pulse width can be further narrowed? If yes, how to realize it?
- (5) If further increasing the irradiation time, whether the mode-locking performance can be improved?
- (6) Whether this defect engineering method is accessible for other low-dimensional materials?

Reviewer #2 (Remarks to the Author):

This manuscript reports interesting novel experimental results of high output mode-locked laser empowered by defect regulation in 2D Bi₂O₂Se saturable absorber, however several points must be revised before being suitable for publication;

1. It is recommended to revise the introduction to introduce the historical review of the saturable absorbers with broadband operation around 2 μm such as MXene Ti₃CNT_x [Adv. Mat., 29, 1702496 (2017)], TMDC WS₂ [Opt. Express., 23, 19996 (2015)], and TI As₂Te₃ [Sci. Rep., 10, 15305 (2020)], since it is a very hot issue for the saturable absorbers to operate at long wavelengths.
2. The authors claim that 2D Bi₂O₂Se has an ultra-broad band response but no data for mode locking at wavelength over 1 μm can be found. The authors are recommended to show data for 1.5 and/or 2 μm operation and compare the results with previous studies using other 2D materials.
3. The pulse widths of the mode-locked laser must be measured together with the spectral width so that the time-bandwidth product can be obtained to ensure proper achievement of well mode-locked pulses without chirping or phase modulations. [Refer to Adv. Mat., 29, 1702496 (2017)]
4. The authors claim that 2D Bi₂O₂Se has excellent air stability but no data for long term operation in open air condition can be found. The authors are recommended to make long term operation of their 2D Bi₂O₂Se saturable absorber and show that the deterioration is negligible under photo-degradation and/or air environment.

Reviewer #3 (Remarks to the Author):

The manuscript by Liu et al. reported a 1.04 μm passively mode-locked Yb:KYW solid-state laser using a defect-regulated 2D Bi₂O₂Se saturable absorber. By inducing and optimizing the defect regulation of the 2D Bi₂O₂Se, they obtained the improved performance of saturable absorption of Bi₂O₂Se, and further generated higher-performance output of the mode-locked laser using the defect-regulated Bi₂O₂Se. Although the authors performed these experiments, I have to say that this work could be not novel and no significant contribution to the fields of nonlinear optics and ultrafast laser, and the reasons are as follows.

1) 2D-material-based saturable absorber (graphene) has been discovered and used to passively mode-lock fiber laser as early as in 2009 [T.Hasan et al., Adv. Mater. 21, 3874, 2009//Q. Bao et al. Adv. Mater. Funct. 19, 3077, 2009], and the 2D Bi₂O₂Se has been previously found as an ultrabroadband saturable absorber, so it is not surprising to obtain the mode-locked laser using the 2D Bi₂O₂Se SA.

2) Defect regulation of mediation is a common way to improve the saturable absorption performance of 2D material, so the use of defect regulated 2D Bi₂O₂Se is also not novel.

3) As seen in Fig.4c, the output power (665 mW) and the pulse duration (266 fs) in this work are also not highest level in the 2D-material mode-lock laser. In Refs.[41, 11], the mode-locked pulse duration can be even as short as 30 fs, and in Refs.[44, 48] the output power is as high as 800 mW.

In summary, this work is not novel in principle and the output laser performance is not surprising. I can not recommend it for publication in the high-level journal Nat. Commun.

Point-by-point responses to the reviewers' comments of the manuscript "*High output mode-locked laser empowered by defect regulation in 2D Bi₂O₂Se saturable absorber*".

Thank you very much for the reviewers' keen interest and constructive comments on our manuscript entitled "*High output mode-locked laser empowered by defect regulation in 2D Bi₂O₂Se saturable absorber*". These comments are all valuable and very helpful for revising and improving our manuscript, as well as making important guiding significance to our research. We have carefully studied the comments and thoroughly made the revisions. The point-by-point responses to the reviewers' comments are listed as follows:

Reviewers' comments and our responses:

Reviewer #1:

This work demonstrates the availability of mode-locking performance control using Bi₂O₂Se nanoplates as a saturable absorber (SA) via defect engineering (i.e., plasma irradiation), which enables tailoring of the saturation intensity and carrier lifetime. An improved output power and narrowed pulse width of mode locking have been achieved based on the optimized SA. Detailed characterizations of both the material and laser have also been performed for intuitive comparisons. The approach proposed in this paper is new and should be quite valuable for the development of ultrafast laser fields. In my option, this work can be considered to be published, after addressing the following issues:

Response: We thank the reviewer for his/her positive comments.

1. *The achieved mode-locked pulses are not operated at soliton state, why sech² instead of Gauss profile was selected to fit the autocorrelation trace?*

Response: We thank the reviewer very much for this kind suggestion. The achieved mode-locking pulses are not operated at the soliton state and are slightly chirped, so the autocorrelation trace should be fitted by Gauss rather than the sech² model. As shown

in Fig. R1, under different plasma irradiation times (0 min, 2 min, 3 min, 5 min) treated on the Bi₂O₂Se nanoplates, the pulse durations were measured to be 661 fs, 591 fs, 400 fs and 312 fs by Gaussian pulse shape fitting. We have revised and updated it in the manuscript (highlighted in yellow, Fig. 1b and Fig. 4a).

Fig. R1 Autocorrelation traces of mode-locking pulses with Bi₂O₂Se nanoplates under different plasma irradiation times. **a** pristine. **b** 2 min. **c** 3 min. **d** 5 min.

2. *How about the long-term stability of the mode-locked laser?*

Response: We thank the reviewer very much for the good suggestion. By using the same Bi₂O₂Se nanoplates under a plasma irradiation time of 5 min, we reperformed the mode-locked experiment and measured the corresponding output power instability and pulse width. As shown in Fig. R2a, the maximum output power was 654 mW obtained three months later, and the instabilities (average output power, rms) of mode-locked operation were measured to be 1.69% at 12 hours. This not only indicates the long-term stability of the mode-locked laser operation but also illustrates the long-term stability of the Bi₂O₂Se nanoplates. In addition, the characterization of CWML operation (Fig. R2b-c) for the same Bi₂O₂Se SA before and after exposure to air for three months demonstrates that the Bi₂O₂Se nanoplate-based SA has excellent long-term stability. We

have revised and updated it in the manuscript (highlighted in yellow, page 11, line 218 and Supplementary Fig. 9).

Fig. R2 a The stability measurements of mode-locked operations based on Bi₂O₂Se nanoplate SA. The average output power **b** and pulse width **c** of the CWML operation based on Bi₂O₂Se SA before and after exposure to air for three months.

3. What limit further power scaling of the mode-locked pulses? Damage of the SA, disappearance of mode-locking, or others? The relative comments should be provided.

Response: We thank the reviewer very much for pointing out this. In our experiment, no optical damage was observed in the Bi₂O₂Se SA during the mode-locked operation. Upon further increasing the pump power, the Bi₂O₂Se SA is completely bleached, and there is insufficient further modulation to drive the pulse forming process. Therefore, the CWML operation would become unstable and eventually disappear under high pump power. We have added the corresponding comments to the revised manuscript (highlighted in yellow, page 10, line 215).

4. *Following the above question, whether the mode-locking pulse width can be further narrowed? If yes, how to realize it.*

Response: We thank the reviewer very much for pointing out this. The saturable absorption parameters (saturation intensity, recovery time, etc.) of SA are one of the factors that determine the mode-locked pulse width. Therefore, defect engineering is employed in Bi₂O₂Se SA by moderating Ar⁺ plasma treatments, which significantly accelerates the trapping rate and defect-assisted Auger recombination rate of photocarriers and empowers strong saturable absorption and self-defocusing properties in Bi₂O₂Se. The improved saturation intensity and ultrafast carrier lifetime make synergetic contributions to the high-performance mode-locked laser with both high output and ultrashort pulse width. The CWML operation was not achieved when further increasing the irradiation time on the Bi₂O₂Se SA to 6 minutes (*The detailed explanation can be found in the response to Question 5*). **In addition to the saturable absorption parameters of SA, the mode-locked pulse width is also determined by the fluorescence linewidth (emission gain linewidth) of the laser crystal, the intracavity dispersion and the energy.** Therefore, by applying a high-quality low-loss cavity mirror (larger coating wavelength band with higher reflection/transmission) and optimizing the dispersion compensation and cavity designation, the mode-locking pulse width can be further narrowed.

5. *If further increasing the irradiation time, whether the mode-locking performance can be improved?*

Response: We thank the reviewer very much for this kind suggestion. As shown in Fig. R3, when increasing the irradiation time on the Bi₂O₂Se nanoplates to 6 minutes, the saturation recovery time of Bi₂O₂Se nanoplates shows an accelerated trend, and the saturation intensity is increased, which is consistent with the phenomenon observed in the manuscript. It seems that the mode-locking performance should be improved by further increasing the irradiation time. However, we did not achieve CWML operation based on the Bi₂O₂Se nanoplate SA treated by 6 min plasma irradiation. First, the larger saturation intensity increases the difficulty of the mode-locking operation. Second, the

introduction of a large number of defect states reduces the material quality, as well as the damage threshold of the Bi₂O₂Se nanoplate.

Fig. R3 a Normalized differential transmission of Bi₂O₂Se nanoplates with 6 min plasma treatment. **b**

Nonlinear transmittance of Bi₂O₂Se nanoplates under 6 min plasma irradiation.

6. *Whether this defect engineering method is accessible for other low-dimensional materials?*

Response: We thank the reviewer very much for pointing out this. Recently, plasma treatment has become an effective method to create atomics defects in 2D materials^{1, 2, 3, 4}, including oxygen plasma⁴ and argon plasma⁵. For example, oxygen plasma treatment of ReS₂ introduced S vacancies and improved its electrical property and photodetection performance¹. Argon plasma generated Se vacancies PdSe₂ to mediate the phase transition of Pd₁₇Se₁₅² and Se vacancies in PtSe₂ to result in thickness-independent semiconducting to metallic conversion³. In our manuscript, we prove that the defect regulation of Bi₂O₂Se nanoplates by oxygen plasma can effectively accelerate their carrier recombination and greatly improve their saturation intensity, thus improving their mode-locked laser performance. Therefore, we believe that this defect engineering method can control the ultrafast carrier dynamics and nonlinear absorption properties of other two-dimensional materials to enhance their pulsed laser performance.

R1. J. Shim. Et. al. High-performance 2D rhenium disulfide (ReS₂) transistors and photodetectors by oxygen plasma treatment. *Adv. Mater.* **28**, 6985 (2016).

-
- R2. Oyedele, A. et. al. Defect-mediated phase transformation in anisotropic two-dimensional PdSe₂ crystals for seamless electrical contacts. *J. Am. Chem. Soc.* **141**, 8928-8936 (2019).
- R3. Shawkat, M. et. al. Thickness-independent semiconducting-to-metallic conversion in wafer-scale two-dimensional PtSe₂ layers by plasma-driven chalcogen defect engineering. *ACS Appl. Mater. Inter.* **12**, 14341-14351 (2020).
- R4. Bhimanapati, G. et. al. Recent advances in two-dimensional materials beyond graphene. *ACS Nano* **9**, 11509-11539 (2015).
- R5. Zhu, J. et. al. Argon plasma induced phase transition in monolayer MoS₂. *J. Am. Chem. Soc.* **139**, 10216-10219 (2017).

Reviewer #2:

This manuscript reports interesting novel experimental results of high output mode-locked laser empowered by defect regulation in 2D Bi₂O₂Se saturable absorber, however several points must be revised before being suitable for publication;

Response: Many thanks for the reviewer's positive comments.

1. *It is recommended to revise the introduction to introduce the historical review of the saturable absorbers with broadband operation around 2 μm such as MXene Ti₃CNT_x [Adv. Mat., 29, 1702496 (2017)], TMDC WS₂ [Opt. Express., 23, 19996 (2015)], and TlAs₂Te₃ [Sci. Rep., 10, 15305 (2020)], since it is a very hot issue for the saturable absorbers to operate at long wavelengths.*

Response: We thank the reviewer for this kind suggestion. We agree with the reviewer that it is a very hot issue for saturable absorbers to operate at long wavelengths. We have added the corresponding description of the above SAs in the revised manuscript (highlighted in yellow, page 2, line 36). Relevant references have also been added to the revised manuscript (refs. 12-14), as shown below:

"12 Jung, M. et. al. 1.94 μm, all-fiberized laser using WS₂-based evanescent field interaction. *Opt. Express* **23**, 19996-20006 (2015).

13 Jhon, Y. I. et. al. Metallic MXene saturable absorber for femtosecond mode-locked lasers. *Adv. Mater.* **29**, 1702496 (2017).

14 Lee, J. et. al. Nonlinear optical properties of arsenic telluride and its use in ultrafast fiber lasers. *Scientific Reports* **10**, 15305 (2020).”

2. *The authors claim that 2D Bi₂O₂Se has an ultra-broad band response but no data for mode locking at wavelength over 1 μm can be found. The authors are recommended to show data for 1.5 and/or 2 μm operation and compare the results with previous studies using other 2D materials.*

Response: We thank the reviewer for this kind suggestion. We agree with the reviewer that adding the data for the mode-locking operation over 1 μm can absolutely illustrate the ultrabroad band response of 2D Bi₂O₂Se and further improve our manuscript. By employing the Bi₂O₂Se SA treated with argon plasma for 5 min, we realized mode-locked laser operation at 2.0 μm to further study its broadband response. The experimental setup is schematically shown in Supplementary Fig. 2a. A 3 × 3 × 8 mm³ Tm:YAP crystal with 4% at. concentration was used as the gain medium. A fiber coupled laser diode emitting at 790 nm with a core diameter of 200 μm and a numerical aperture of 0.22 was used as the pump source. With a 1.8:1 optical collimation system, the pump spot radius focused into the crystal was ~56 μm, which was well matched with the radius of the TEM₀₀ cavity mode (55 μm) calculated by ABCD propagation matrix theory. The oscillation laser mode radius on Bi₂O₂Se SA was 38 μm. The dichroic mirrors M1 (R=∞), M2 (R=0.8 m), and M3 (R=0.1 m) were all HR coated at 1.8-2.1 μm and high transmission (HT) coated at 780–810 nm. A flat output coupler (OC) with a transmission of 1% for a spectral range of 1.8-2.1 μm was used. Two Gires–Tounois interferometer (GTI) mirrors with a total GDD of –600 fs² per round were used to compensate for the normal dispersion introduced by the crystal and Bi₂O₂Se SA.

With the Bi₂O₂Se SA used in the cavity, after careful adjustment, the laser runs into a continuous wave mode-locked (CWML) regime when the absorbed pump power exceeds 5.53 W, as shown in Fig. R4a. Under an absorbed pump power of 6.22 W, a

maximum average output power of 51 mW is obtained. As shown in Fig. R4b, the output mode-locked laser spectrum is centered at 1941 nm with a full-width at half-maximum (FWHM) of 6.7 nm. Fig. R4c shows the pulse trains on the 200 ns and 500 μ s time scales at the maximum output power, which indicates the realization of mode-locked laser operation. In addition, the recorded radio frequency spectrum is shown in Fig. R4d, with a fundamental beat note near 42.8 MHz and the corresponding signal-to-noise ratio of 52 dB, which is measured by a spectrum analyzer (Agilent N9000A) with a resolution bandwidth (RBW) of 5 kHz. The inset of Fig. R4d is recorded over a wide span of 1 GHz with an RBW of 1.0 MHz, and the absence of any spurious modulation proves clean CWML operation at 2.0 μ m based on Bi₂O₂Se SA. Due to the limitation of the measurement device (autocorrelator: Pulse Check 150) and at 2.0 μ m, the mode-locked pulse duration is not measured, which can be deduced from the mode-locked FWHM (4.4 nm) and the time bandwidth product (0.441) to be \sim 1.25 ps. From the summarization of the mode-locked laser operation at 2.0 μ m based on 2D materials (Table R1), it is obvious that the pulsed laser with Bi₂O₂Se nanoplates has a relatively shorter pulse width and higher average output power. In conclusion, the results of our experiments confirm that Bi₂O₂Se nanoplates can be used as an effective broadband saturable absorption material for pulse generation at 2 μ m wavelengths. We have carefully revised and updated it in the manuscript (highlighted in yellow, page 11, line 224). The corresponding results have been added to Supplementary Fig. 10 and Supplementary Table 3 in the Supplementary Information.

Fig. R4 Mode-locked laser based on Bi₂O₂Se nanoplates treated with argon plasma for 5 min at 2.0 μm. **a** Average output power of the CWML versus absorbed pump power. **b** Output mode-locked spectrum. **c** The pulse train with a span of 200 ns and 500 μs. **d** Recorded frequency spectrum with wide and narrow (inset) spans.

Table R1. Results of mode-locked lasers operating at 2.0 μm based on 2D materials

Materials	Laser Gain Materials	Pulse Width (fs)	Output Power (mW)	Reference
Graphene	Tm ³⁺ :CLNGG	729	60	6
SWCNTs	Tm ³⁺ :Lu ₂ O ₃	175	36	7
BP	Tm ³⁺ silica fiber	793	1.5	8
MoS ₂	Tm ³⁺ silica fiber	1510	8	9
MoTe ₂	Tm ³⁺ silica fiber	952	36.7	10
Ti ₃ C ₂ T _x	Tm ³⁺ -Ho ³⁺ -codoped fiber	897	12.5	11
V ₂ C	Tm ³⁺ -Ho ³⁺ -codoped fiber	843	14	12
Bi ₂ O ₂ Se	Tm ³⁺ :YAP	~1250	51	This work

-
- R6. Ma, J. et. al. Graphene mode-locked femtosecond laser at 2 \$\mu\text{m}\$ wavelength. *Opt. Lett.* **37**, 2085-2087 (2012).
- R7. Schmidt, J. et. al. 175 fs Tm:Lu₂O₃ laser at 2.07 \$\mu\text{m}\$ mode-locked using single-walled carbon nanotubes. *Opt. Express* **20**, 5313-5318 (2012).
- R8. Sotor, J., Sobon, G., Kowalczyk, M., Macherzynski, W., Paletko, P., Abramski, KM., Ultrafast thulium-doped fiber laser mode locked with black phosphorus. *Opt. Lett.* **40**, 3885-3888 (2015).
- R9. Cao, L. et. al. Tm-doped fiber laser mode-locking with MoS₂-polyvinyl alcohol saturable absorber. *Opt. Fiber Technol.* **41**, 187-192 (2018).
- R10. Wang, J. et. al. Mode-locked thulium-doped fiber laser with chemical vapor deposited molybdenum ditelluride. *Opt. Lett.* **43**, 1998-2001 (2018).
- R11. Jhon, Y., Lee J, Jhon, Y., Lee, J., Ultrafast mode-locking in highly stacked Ti₃C₂Tx MXenes for 1.9 \$\mu\text{m}\$ infrared femtosecond pulsed lasers. *Nanophotonics* **10**, 1741-1751 (2021).
- R12. Lee, J., Kwon, S., Lee, J., Investigation on the nonlinear optical properties of V₂C MXene at 1.9 \$\mu\text{m}\$. *J. Mater. Chem. C*, **9**, 15346-15353 (2021).
3. *The pulse widths of the mode-locked laser must be measured together with the spectral width so that the time-bandwidth product can be obtained to ensure proper achievement of well mode-locked pulses without chirping or phase modulations. [Refer to Adv. Mat., 29, 1702496 (2017)]*

Response: We thank the reviewer for this kind suggestion. We agree with the reviewer that the pulse widths of the mode-locked laser must be measured together with the spectral width. As shown in Fig. R5, the spectral widths of a CWML laser based on Bi₂O₂Se nanoplates under different plasma irradiation times are 2.5 nm, 2.8 nm, 4.1 nm and 5.3 nm. Their corresponding time bandwidths are ~0.452, 0.453, 0.454 and 0.452, respectively, which indicates that the output pulse is slightly chirped. The corresponding spectral width and comments have been added to the revised manuscript (highlighted in yellow, page 10, line 206 and Supplementary Fig. 2b, 8a-c).

Fig. R5 The spectrum of the mode-locked operation based on $\text{Bi}_2\text{O}_2\text{Se}$ nanoplates under different plasma irradiation times. **a** Pristine. **b** 2 min. **c** 3 min. **d** 5 min.

4. The authors claim that 2D $\text{Bi}_2\text{O}_2\text{Se}$ has excellent air stability, but no data for long-term operation in open air condition can be found. The authors are recommended to make long term operation of their 2D $\text{Bi}_2\text{O}_2\text{Se}$ saturable absorber and show that the deterioration is negligible under photo-degradation and/or air environment

Response: Thank you very much for the kind suggestion. As shown in Fig. R6, the characterization of the Raman spectra (Fig. R6a) and CWML operation (Fig. R6b-c) for the same $\text{Bi}_2\text{O}_2\text{Se}$ SA before and after exposure to air for three months demonstrate that the device has excellent long-term stability. Moreover, the instabilities (average output power, rms) of the mode-locked operation are measured to be 1.69% at 12 h. The corresponding information has been revised and updated in the manuscript (highlighted in yellow, page 11, line 218 and Supplementary Fig. 1d, 9a-c).

Fig. R6 a Raman spectra of the Bi₂O₂Se nanoplates before and after exposure to air for three months. b The average output power and c pulse width of the CWML operation based on Bi₂O₂Se SA before and after exposure to air for one month. d The stability measurements of mode-locked operations based on Bi₂O₂Se nanoplate SA.

Reviewer 3:

The manuscript by Liu et al. reported a 1.04 μm passively mode-locked Yb:KYW solid-state laser using a defect-regulated 2D Bi₂O₂Se saturable absorber. By inducing and optimizing the defect regulation of the 2D Bi₂O₂Se, they obtained the improved performance of saturable absorption of Bi₂O₂Se and further generated higher-performance output of the mode-locked laser using the defect-regulated Bi₂O₂Se. Although the authors performed these experiments, I have to say that this work could be not novel and no significant contribution to the fields of nonlinear optics and ultrafast laser, and the reasons are as follows.

Response: Many thanks to the reviewer for reviewing our manuscript and giving his/her constructive comments.

1. *2D-material-based saturable absorber (graphene) has been discovered and used to passively mode-lock fiber laser as early as in 2009 [T. Hasan et al., Adv. Mater. 21, 3874, 2009//Q. Bao et al. Adv. Mater. Funct. 19, 3077, 2009], and the 2D Bi₂O₂Se has been previously found as an ultrabroadband saturable absorber, so it is not surprising to obtain the mode-locked laser using the 2D Bi₂O₂Se SA.*

Response: Graphene was proven to be an excellent saturable absorber as early as 2009^{13, 14} (we have made contributions to the early work, Q. Bao et al. Adv. Mater. Funct. 19, 3077, 2009), which opened a door and became a hot research topic for exploring the nonlinear optical properties of 2D materials as well as their applications as SA for passive Q-switching and mode-locking laser operation. Since then, a variety of 2D materials, including but limited to transition metal dichalcogenides (TMDs; e.g., MoS₂ and WS₂), black phosphorous (BP), topological insulators (TIs; e.g., Bi₂Te₃ and Sb₂Se₃), MXenes, and 2D material-based heterostructures, have been studied for pulsed laser generation. However, each type of the above material has its drawbacks: for graphene, the weak absorptance (single layer: ~2.3%) and limited density of states in graphene are obstacles to achieving a high modulation depth; for TMDs, the extremely high saturation intensity of TMDs on the order of tens of GW/cm² makes continuous-wave mode-locked (CWML) operation difficult in all-solid-state lasers, and the large bandgap makes them only to be used in visible and near infrared regimes. For BP, poor air stability is the largest obstacle for its real applications; for TIs, poor air compatibility and relatively low modulation depth limit its application in high-power high-energy pulsed lasers. Therefore, it is urgent to continue exploring novel 2D materials as well as the properties and performance modulation techniques. Moreover, due to the limited density of states and extremely high saturation intensity of common 2D materials (e.g., graphene and TMDs), it is difficult to generate ultrafast lasers with both high output power and ultrashort pulse width simultaneously.

Recently, 2D bismuth oxyselenide (Bi₂O₂Se) has been demonstrated to have ultrabroadband nonlinear modulation (from 1.55 μm to 5.0 μm) and a larger nonlinear absorption coefficient β (-2.91×10^{-6} cm/W @800 nm. Although it has been

demonstrated to have saturable absorption¹⁵ (this work was based on Bi₂O₂Se nanoflakes produced by a solution method wherein many chemicals were used, which may have contamination effects on the optical property investigations), due to the low saturation intensity of Bi₂O₂Se (~ MW/cm²), ultrafast mode-locked lasers based on Bi₂O₂Se SA have not been devised. In this work, for the first time, to the best of our knowledge, a femtosecond solid-state laser with 2D Bi₂O₂Se nanoplates as SA is realized. **Both a high output power of ~665 mW and an ultrashort pulse width of ~312 fs are achieved upon defect engineering in Bi₂O₂Se SA.** Moreover, defect engineering is triggered by the regulation of the O and Se defect states in Bi₂O₂Se nanoplates via moderate Ar⁺ plasma treatments, which significantly accelerates the trapping rate and defect-assisted Auger recombination rate of photocarriers and empowers strong saturable absorption and self-defocusing properties in Bi₂O₂Se in the NIR region. The improved saturation intensity and ultrafast carrier lifetime make synergetic contributions to the high-performance mode-locked laser with both high output and ultrashort pulse width. **This makes the plasma-treated Bi₂O₂Se a superior 2D SA among the 2D materials in solid-state lasers.**

Overall, our main innovation is illustrating that **Bi₂O₂Se is a superior 2D SA, defect modulation is an effective way to modulate the SA performance, and a high output power and ultrashort pulse mode-locked bulk laser can be realized and improved by plasma-treated Bi₂O₂Se SA.**

R13. Hasan, T. et, al. Nanotube-polymer composites for ultrafast photonics. *Adv. Mater.* **21**, 3874-3899 (2009).

R14. Bao, Q. et, al. Atomic-layer graphene as a saturable absorber for ultrafast pulsed lasers. *Adv. Funct. Mater.* **19**, 3077-3083 (2009).

R15. Tian, X. et, al. An ultrabroadband mid-infrared pulsed optical switch employing solution-processed bismuth oxyselenide. *Adv. Mater.* **30**, 201801021 (2018)

2. *Defect regulation of mediation is a common way to improve the saturable absorption performance of 2D material, so the use of defect-regulated 2D Bi₂O₂Se is also not novel.*

Response: We agree with the reviewer that atomic defects, being the most prevalent zero-dimensional topological defects, are ubiquitous in a wide range of 2D materials. Hence, controlling atomic defects provides an alternative avenue for engineering a wide range of physical and chemical properties of 2D materials to enhance the performance of their corresponding devices. However, there are few reports on enhancing the performance of 2D material-based mode-locked lasers by defect regulation. In this work, defect regulation effectively accelerates carrier recombination and greatly improves the saturation intensity of Bi₂O₂Se SA. The improved saturation intensity and ultrafast carrier lifetime endow the high-performance mode-locked laser with both high output and short pulse duration. In our opinion, the emergence of this work effectively fills the gap in the current field.

3. *As seen in Fig. 4c, the output power (665 mW) and the pulse duration (266 fs) in this work are also not highest level in the 2D-material mode-lock laser. In Refs. [41, 11], the mode-locked pulse duration can be even as short as 30 fs, and in Refs. [44, 48] the output power is as high as 800 mW.*

Response: It is certainly that the output power is not the highest, and the pulse width is not the shortest result among 2D material-based mode-locked lasers. In Refs. [Opt. Lett. 41, 890-893 (2016)], although the mode-locked pulse duration can be as short as 30 fs, its average power is only 26 mW. In Refs. [Opt. Lett. 38, 4189-4192 (2013)], the output power is as high as 800 mW, while the mode-locked pulse duration has reached 643 fs. Therefore, it is a great challenge to realize mode-locked lasers with high average output power and narrow pulse width at the same time. The main novelty of our work is to explore a new material and a novel performance modulation technique to realize ultrafast lasers with both high output power (665 mW) and an ultrashort pulse width (312 fs) simultaneously.

REVIEWER COMMENTS

Reviewer #1 (Remarks to the Author):

The authors have taken all reviewer points in consideration, with clear, detailed, and reasoned responses to all. The corresponding changes in the manuscript address these concerns appropriately. In this current form this manuscript should be considered acceptable for publication.

Reviewer #2 (Remarks to the Author):

The authors claim that the pulse shape is not sech² but Gaussian just because it is chirped, which is very absurd. Properly mode-locked soliton pulses have nearly sech² shape even though they are chirped and must be fitted as such. The time-bandwidth product (TBP) is 0.32 for sech² transformed-limited pulses and TBP>0.44 implies severe phase modulation due to substantial chirping. The authors are recommended to measure and control the dispersion of each component in the laser cavity so that the dispersion of the laser cavity is optimized to obtain proper mode locking.

Reviewer #3 (Remarks to the Author):

I have carefully read the revised manuscript and the response letter, and the authors have well addressed all of my questions. Although this work is not the first report on 2D-material-based mode-locked femtosecond laser, the defect-regulated 2D Bi₂O₂Se saturable absorber (SA) proposed by the authors indeed shows the superior overall performance compared with other 2D-material SAs (e.g. graphene, TMD, TI, etc). Namely, it is very important that such Bi₂O₂Se SA can provide both of a large modulation depth and low saturable intensity, which is favorable to the high-power ultrafast laser generation. Moreover, it is impressive in the revised manuscript that they tested the stability of the ultrafast laser during three months.

In a word, they give the convincing explanation to my questions, and this work may be an important step for 2D-material mode-locked high-power laser into practical application. Therefore, I agree to accept the revised manuscript for publication in Nature Communication now.

Point-by-point responses to the reviewers' comments of the manuscript "*High output mode-locked laser empowered by defect regulation in 2D Bi₂O₂Se saturable absorber*".

We sincerely appreciate the reviewers' keen interest and constructive comments on our manuscript entitled "*High output mode-locked laser empowered by defect regulation in 2D Bi₂O₂Se saturable absorber*". The comments are all valuable and helpful for revising and improving our manuscript, as well as making important guiding significance to our research. We have carefully studied the comments and thoroughly made the revisions. The point-by-point responses to the reviewers' comments are listed as follows:

Reviewers' comments and our responses:

Reviewer #1:

The authors have taken all reviewer points in consideration, with clear, detailed, and reasoned responses to all. The corresponding changes in the manuscript address these concerns appropriately. In this current form this manuscript should be considered acceptable for publication.

Response: Many thanks for the reviewer's positive comments and appreciation of our work, especially the constructive recommendation and helpful advice.

Reviewer #2:

The authors claim that the pulse shape is not sech^2 but Gaussian just because it is chirped, which is very absurd. Properly mode-locked soliton pulses have nearly sech^2 shape even though they are chirped and must be fitted as such. The time-bandwidth product (TBP) is 0.32 for sech^2 transformed-limited pulses and $\text{TBP} > 0.44$ implies severe phase modulation due to substantial chirping. The authors are recommended to measure and control the dispersion of each component in the laser cavity so that the dispersion of the laser cavity is optimized to obtain proper mode locking.

Response: We thank the reviewer very much for pointing out this important and constructive comment, which can significantly improve our manuscript.

In real mode-locked lasers, the pulse shape is complicated since dispersion management can produce many different pulse shapes: (i) sech² pulses with a small dispersion swing; (ii) Gaussian pulses with moderate dispersion; and (iii) the pulse shape depends more sensitively on gain filtering with large dispersion [1]. In addition, for passive mode locking based on a slow saturable absorber with self-phase modulation (SPM) and group delay dispersion (GDD), the pulse shape should be sech² if there is soliton formation, while it should be a Gaussian shape if there is no soliton formation [2].

We apologize for the confusing description of the obtained pulse shapes in the previous response letter. We agree with the reviewer that mode-locking soliton pulses have a nearly sech² shape even though they are chirped and must be fitted as such. According to the reviewer's suggestion, **we have carefully analyzed our results (especially the autocorrelation curve and spectrum of the mode-locking pulses), reconsidered the dispersion of the laser cavity, and finally concluded that the achieved mode-locked pulses in our experiment should be sech² shaped.**

First, it is a passive mode-locking operation with a slow saturable absorber since the obtained mode-locking pulse duration is 10 times shorter than the recovery time of the 2D Bi₂O₂Se saturable absorber.

Second, two Gires-Tournois interferometer (GTI) mirrors with a total negative GDD of ~-750 fs² per round were used to compensate for the mirror dispersion (low-loss chirped mirrors with near-zero GDD) and material dispersion (mainly derived from the laser crystal with positive GDD ~700 fs²) inside the resonator and balance the SPM induced by Kerr nonlinearity of the crystal. The total dispersion of the laser cavity was small and negative.

Third, as shown in Fig. R1-R2, the autocorrelation trace and spectrum of mode-locked pulses are both almost perfectly sech² fitted, which is evidence for soliton-like

pulse generation (although in theory, in both cases, this is an approximation) [3]. As shown in Fig. R1-R2, with Bi₂O₂Se nanoplates under different plasma irradiation times (0 min, 2 min, 3 min, 5 min), the pulse durations obtained are determined to be 587 fs, 454 fs, 340 fs and 266 fs, respectively, with sech² fitting. The spectrum bandwidths (FWHM) fitted by sech² are 2.1 nm, 2.8 nm, 3.8 nm and 4.8 nm. Therefore, **the corresponding time-bandwidth product (TBP) is recalculated to be 0.337, 0.348, 0.353 and 0.349, which are slightly higher than that of the transformed-limited sech² shape (0.315).**

In conclusion, (i) the dispersion of our laser cavity is small and negative; (ii) the autocorrelation trace and spectrum of mode-locked pulses both are well described by sech² fitting; (iii) the TBP of obtained mode-locking pulses with sech² shapes is slightly higher than that of the transformed limitation and indicates slight chirping; and (iv) limited by the elements in hand, the dispersion of the cavity has been optimized for proper mode locking with long-term stability (rms=1.69% @12 hrs).

Based on the above experimental evidence and analysis, **we are confident that the mode-locking pulses obtained in our experiment are sech² shapes.**

We have carefully revised and updated it in the revised manuscript (highlighted in yellow, page 4, line 77, page 10, line 206, Fig. 1b, Fig. 4a, Fig. 4c, Supplementary Fig. 2b and Supplementary Fig. 8).

Fig. R1 Autocorrelation traces of mode-locking pulses with Bi₂O₂Se nanoplates under different plasma irradiation times. a pristine. b 2 min. c 3 min. d 5 min.

Fig. R2 The spectrum of the mode-locked pulses based on $\text{Bi}_2\text{O}_3\text{Se}$ nanoplates under different plasma irradiation times. a Pristine. b 2 min. c 3 min. d 5 min.

R1. Haus, H. A. et, al. Dispersion-managed mode locking. *J. Opt. Soc. Am. B*, **16**, 1999-2004 (1999).

R2. Keller, U. Passive Mode-locking. *Ultrafast Lasers: A Comprehensive Introduction to Fundamental Principles with Practical Applications. Springer International Publishing: Cham* **9**, 419-546 (2021).

R3. Huang, Z. et, al. SESAM mode-locked Yb:SrLaAlO₄ laser. *Opt. Express*, **29**, 43820-43826 (2021).

Acknowledgments

We sincerely appreciate Prof. Chaokuei Lee from National Sun Yat-Sen University, Prof. Jie Ma from Jiangsu Normal University and Prof. Weidong Chen from Fujian

Institute of Research on the Structure of Matter for discussing the obtained mode-locking pulse shapes.

Reviewer 3:

I have carefully read the revised manuscript and the response letter, and the authors have well addressed all of my questions. Although this work is not the first report on 2D-material-based mode-locked femtosecond laser, the defect-regulated 2D Bi₂O₂Se saturable absorber (SA) proposed by the authors indeed shows the superior overall performance compared with other 2D-material SAs (e.g., graphene, TMD, TI, etc.). Namely, it is very important that such Bi₂O₂Se SA can provide both of a large modulation depth and low saturable intensity, which is favorable to the high-power ultrafast laser generation. Moreover, it is impressive in the revised manuscript that they tested the stability of the ultrafast laser during three months.

In a word, they give the convincing explanation to my questions, and this work may be an important step for 2D-material mode-locked high-power laser into practical application. Therefore, I agree to accept the revised manuscript for publication in Nature Communication now.

Response: Many thanks for the reviewer's positive remarks, which certainly confirm the novelty and importance of our work.

REVIEWERS' COMMENTS

Reviewer #2 (Remarks to the Author):

The authors have recognized their mistake and agreed that the mode-locked pulses are sech² shape and well revised their entire manuscript as such, which significantly improved not only the credibility but also the quality of the manuscript.

However, In Fig. 4 a, the autocorrelation trace for 2 min before revision is very noisy and asymmetric, while it became very clean after revision without any explanations. The authors must explain the reason and clarify the process.

Point-by-point responses to the reviewers' comments of the manuscript "*High output mode-locked laser empowered by defect regulation in 2D Bi₂O₂Se saturable absorber*".

We sincerely appreciate the reviewers' keen interest and constructive comments on our manuscript entitled "*High output mode-locked laser empowered by defect regulation in 2D Bi₂O₂Se saturable absorber*". The comments are all valuable and helpful for revising and improving our manuscript, as well as making important guiding significance to our research. We have carefully studied the comments and thoroughly made the revisions. The point-by-point responses to the reviewers' comments are listed as follows:

Reviewers' comments and our responses:

Reviewer #2:

The authors have recognized their mistake and agreed that the mode-locked pulses are sech² shape and well revised their entire manuscript as such, which significantly improved not only the credibility but also the quality of the manuscript. However, In Fig. 4a, the autocorrelation trace for 2 min before revision is very noisy and asymmetric, while it became very clean after revision without any explanations. The authors must explain the reason and clarify the process.

Response: We thank the reviewer very much for pointing out this. We are sorry for forgetting to give the corresponding explanation about the revision of Fig. 4a in the last response letter. According to the reviewer's previous suggestion, we have carefully analyzed our results and confirmed the dispersion of the laser cavity and finally demonstrated the conclusion by redoing the mode-locking experiment with the Bi₂O₂Se SA treated with argon plasma for 2 mins under the same configuration of the previous experiment. The pulse profile obtained was cleaner and more symmetrical, while the spectrum and output power were almost the same as the previous results, so we modified Fig. 4a.